# Industry 4.0 readiness and strategic plan failures in SMEs: A comprehensive analysis

**Umawathy Techanamurthy[1], Muhammad Saqib Iqbal [2]\* Zulhasni Abdul Rahim[3]**

**1** Department of Engineering Education, Faculty of Engineering Education & Built Environment, National University of Malaysia, Bangi, Selangor, Malaysia, **2** NUST Business School, National University of Science and Technology, Islamabad, Pakistan, **3** Malaysia-Japan International Institute of Technology (MJIIT), University Teknologi Malaysia (UTM), Wilayah Persekutuan Kuala Lumpur, Malaysia

\* saqib.iqbal@nbs.nust.edu.pk

## Abstract

This study examines the industry 4.0 readiness of 506 Malaysian SMEs, focussing on key factors that contribute to frequent failures in their strategic planning processes. Driven by rapid technological advancements such as IoT, AI, and big data analytics, Industry 4.0 presents opportunities and challenges for SMEs striving to remain competitive in a global digital economy. Despite government efforts such as Malaysia's Industry4WRD programme, SMEs need help with leadership, digital infrastructure, and workforce competency. This study aims to provide an in-depth analysis of these readiness factors using data from the Industry4WRD assessment framework, evaluating dimensions such as leadership, governance, digital infrastructure, workforce competency, and strategic alignment on a scale of 0–4. The results indicate that leadership and strategic alignment score the highest with a mean of 0.81, while workforce competency scores the lowest at 0.35, highlighting significant competency gaps. Leadership shows a strong positive correlation (0.92) with overall readiness, underscoring its critical role. Sectoral analysis reveals that the chemical sector demonstrates the highest readiness, while Selangor and Johor are the leading regions driven by more robust digital infrastructure. This study emphasises the need for targeted improvements in leadership development, digital infrastructure, and workforce training to bridge readiness gaps, particularly in less developed regions and industries. The findings support the objectives of Malaysia's New Industrial Master Plan 2030 (NIMP 2030), offering actionable insights for policy makers and leaders of SMEs with the aim of fostering comprehensive digital transformation and strengthening the competitiveness of SMEs in the era of Industry 4.0.

## 1 Introduction

In today's rapidly evolving digital landscape, Industry 4.0 technologies such as IoT, AI, and big data analytics are reshaping global business processes. Despite

---

**Data availability statement:** The data supporting this study's findings are available upon reasonable request from Sustainable Industrial Revolution and Innovation Sdn Bhd (SIRI), a Universiti Teknologi Malaysia (UTM) spin-off, in compliance with privacy and ethical commitments made to study participants. Due to the sensitive nature of the data and privacy protections outlined in participant consent forms, the data are not publicly accessible. Interested researchers may request data by contacting the SIRI administration at admin@siri.my, referencing the study title and Industry4WRD Readiness Assessment Baseline Report.

**Funding:** The author(s) received no specific funding for this work.

**Competing interests:** The authors have declared that no competing interests exist.

widespread digital transformation initiatives, small and medium enterprises (SMEs) face challenges that hinder their ability to integrate these technologies effectively [1,2]. Prior studies have focused predominantly on larger firms or isolated dimensions of digital adoption, leaving a critical gap in understanding the comprehensive readiness of SMEs. This study aims to bridge that gap by assessing key dimensions, leadership, governance, digital infrastructure, workforce competency, and strategic alignment, in 506 Malaysian SMEs. By examining sectoral and regional disparities, our research provides targeted insight for policy makers and business leaders seeking to enhance SME competitiveness in the Industry 4.0 era.

The advent of Industry 4.0, characterised by the integration of digital technologies such as the Internet of Things (IoT) [3], Artificial Intelligence (AI), and big data analytics, has reshaped global manufacturing and business processes, bringing promising advances in operational efficiency, innovation, and competitiveness [4–6]. However, despite these potential benefits, many SMEs in Malaysia struggle to adopt Industry 4.0 technologies effectively, leading to failures in strategic plans and limitations in their ability to compete in a rapidly evolving digital economy [1,7].

SMEs are a vital component of Malaysia's economy, contributing substantially to employment and GDP, approximately 38% to national GDP and nearly 70% to overall employment. However, barriers such as limited financial resources, inadequate technical expertise, and underdeveloped digital infrastructure hinder their progress toward digital transformation, putting them at risk of lagging behind larger firms and international competitors [7,8]. These challenges risk leaving small businesses behind if not addressed as larger companies and global competitors accelerate their technological transformation.

Recognising the importance of Industry 4.0, Malaysia's New Industrial Master Plan 2023 (NIMP 2030) outlines a comprehensive strategy to accelerate technology adoption in all industries. Specifically, Action Plan 2.1.1 within NIMP 2030 aims to improve the offerings of the Industry4WRD programmeme, including the Readyness Assessment and Intervention Fund, to help SMEs build the digital infrastructure, workforce competency, and leadership skills necessary for the adoption of Industry 4.0. This aligns with the larger goals of NIMP 2030, which aims to position Malaysia as a globally competitive industrial hub by promoting technological advancement and digitalisation.

Given the critical role of SMEs in Malaysia's economic landscape, a thorough assessment of their readiness for Industry 4.0 is essential. This study assessed the readiness of 506 SMEs across Malaysia to adopt Industry 4.0 technologies, focussing on crucial dimensions such as leadership, governance, digital infrastructure, workforce competency, and strategic alignment. Research contributes to the field by identifying the factors that facilitate or hinder Industry 4.0 adoption among SMEs to provide actionable insights for policymakers and business leaders to improve the competitiveness of SMEs. The primary research question driving this study is the following. What are the critical readiness factors for adopting Industry 4.0 technologies among Malaysian SMEs?

The structure of this paper is as follows: the literature review offers a theoretical background on Industry 4.0 and the readiness frameworks; the methodology

describes the sampling, data collection and analysis approach; the results detail the readiness levels in dimensions and the correlation findings; and the discussion provides recommendations for improving digital readiness in Malaysian SMEs.

## 2  Review of the literature

### 2.1  Industry 4.0 concepts

Industry 4.0, introduced by the German government in 2011, has become synonymous with the digital transformation of manufacturing and business processes, promising improved efficiency, productivity, and flexibility [9]. These technologies offer significant competitive advantages for small businesses, including improved operational efficiency and cost reductions. However, SMEs face considerable challenges in adopting Industry 4.0 technologies due to limited financial resources, insufficient technical expertise, and inadequate digital infrastructure [10].

Recent research such as [8,10] underscores that while Industry 4.0 offers significant competitive advantages, SMEs are often constrained by limited resources, underdeveloped digital infrastructure, and skills gaps. Although leadership has been identified as a pivotal factor [11], there remains limited empirical evidence on how leadership interacts with other critical dimensions, such as governance and strategic alignment, in the context of SME. Furthermore, while many studies have addressed isolated aspects of digital readiness, few have employed a comprehensive framework that simultaneously evaluates multiple dimensions. Our study addresses this shortcoming by integrating a multidimensional assessment that measures individual readiness factors and examines their interdependencies, thus contributing a nuanced perspective to the digital transformation literature.

### 2.2  Challenges for SMEs in Industry 4.0 Adoption

Given the importance of Industry 4.0 readiness for competitiveness, various frameworks have emerged to assess enterprise readiness. These frameworks assess vital dimensions such as leadership, governance, digital infrastructure, and workforce competencies. For example, the S-curve analysis [12] and the technology life cycle frameworks assess an organisation's current technological capabilities and identify improvement areas. The TRIZ methodology, as discussed by Iqbal & Rahim in their previous studies [13–16], is widely used for solving complex innovation-related problems and has also been applied to Industry 4.0 readiness, providing a systematic approach to innovation [17].

### 2.3  Industry4WRD Intervention Framework for SMEs

The Industry4WRD intervention framework is a structured initiative to accelerate the adoption of Industry 4.0 technologies in Malaysian industries. Central to the framework are two key components: the readiness assessment and the intervention fund. The readiness assessment assesses the technological readiness of an organisation, while the Intervention Fund provides financial support to address identified gaps in digital infrastructure, workforce competency, and leadership capabilities. This framework plays a crucial role in addressing the unique challenges of Malaysian SMEs, aligning with broader efforts to drive industrial competitiveness through digital transformation [18].

Strategic planning is essential for SMEs to navigate the complex landscape of Industry 4.0. Effective planning involves setting clear objectives, efficiently allocating resources, and continuously monitoring progress. However, many SMEs need better strategic planning due to poor technological integration, inadequate resource management, and a lack of skilled personnel [19]. Research shows that aligning strategic planning with Industry 4.0 requirements can increase the success rate of initiatives by providing clear roadmaps and resource management strategies [20]. This insight is crucial for policy makers and business leaders, as it offers actionable recommendations for enhancing SME performance in Industry 4.0 environments.

Digital transformation is a fundamental element of Industry 4.0 readiness, requiring substantial investments in infrastructure, technology, and workforce competencies. Although digitalisation brings opportunities for SMEs, it also presents

challenges, particularly in developing countries, where digital infrastructures often need to be developed [21]. Workforce competency is critical in the adoption of Industry 4.0, as employees need the necessary skills to use and integrate new technologies in their work processes effectively. This resonates with the scope of the journal, which encourages research that addresses technological advances, workforce innovation, and their integration into business models.

There is considerable variability among sectors and regions regarding Industry 4.0 readiness. Specific industries, such as chemical and electrical equipment, lead to technology adoption, while others, such as manufactured metal products and machinery, lag [8]. Similarly, there are regional disparities, with states that have invested in digital infrastructure demonstrating higher readiness levels [21]. This variability is essential to understand the specific needs of different sectors and regions, contributing to the journal's commitment to inclusive research that addresses diverse economic and geographical contexts.

Several studies have explored the challenges and opportunities associated with Industry 4.0 readiness in SMEs. Ghobakhloo (2020) emphasises the importance of leadership and governance in driving successful digital transformation, while Beier et al. (2021) highlight the role of strategic planning as a determinant of Industry 4.0 success. Similarly, Holzinger et al. (2021) examine how digital transformation contributes to achieving the Sustainable Development Goals (SDGs), particularly in areas such as security, safety, and privacy in a connected world. This focus on sustainability aligns with Cogent Business & Management's encouragement of interdisciplinary research, which bridges innovation and sustainable business practices.

### 2.4 Research gap and study contribution

Although various frameworks and initiatives support Industry 4.0 adoption, there needs to be more understanding of the sectoral and regional disparities in Industry 4.0 readiness among Malaysian SMEs. Most studies focus on general barriers to adoption, but lack critical insights into specific readiness dimensions, particularly leadership and workforce competency, that significantly impact strategic plan success [10,22]. By addressing these readiness dimensions in the context of Malaysian SMEs, this study contributes to the literature by identifying targeted interventions to improve competitiveness, especially in sectors and regions lagging in digital readiness.

## 3 Methodology

This study uses a cross-sectional survey design to evaluate the readiness of 506 Malaysian SMEs for Industry 4.0. Data were obtained from the Industry4WRD programme, a Malaysian government initiative aimed at assessing and enhancing the technological readiness of SMEs to adopt Industry 4.0 technologies such as IoT, AI and big data analytics [23]. SMEs were strategically selected based on firm size, industry classification, and geographic location to ensure a representative sample that captures regional and sectoral differences [22,24].

### 3.1 Technical rigour and data integrity

We implemented rigorous controls throughout the study to ensure that our research is technically sound and that the data robustly support the edconclusions. The survey instrument was developed based on established frameworks, pilot tested, and validated through expert review to ensure reliability and accuracy. A representative sample of 506 SMEs was carefully selected, providing sufficient statistical power and diversity. Additionally, multiple statistical analyses were conducted, including correlation, regression, and Analysis of Variance (ANOVA) to examine the relationships among key dimensions and confirm the reproducibility of our findings. These rigorous experimental and analytical procedures ensure that our conclusions are drawn appropriately on the basis of the data presented.

### 3.2 Data collection and survey instrument

A structured questionnaire was developed to measure five key dimensions of industry 4.0 readiness.

- Leadership: Measured through items that assess top management support, strategic vision, and decision-making processes that guide digital transformation. For example, respondents rate how their leadership prioritises innovation and drives technology adoption on a 0–4 Likert scale.
- Governance: Evaluated by examining the clarity of organisational roles, communication channels, and collaborative processes that support digital initiatives.
- Digital Infrastructure: Assessed using items determining the availability, adoption, and effective use of digital tools, IT systems, and automation technologies within the firm.
- Workforce Competency: Measured by questions about employee skills, training initiatives, and the ability of the workforce to adapt to new technologies.
- Strategic Alignment: This is determined by assessing the consistency between digital initiatives and overall business strategies, including how well IT investments align with long-term corporate objectives.

Each element was rated on a standardised 0–4 Likert scale, where higher scores indicate greater readiness. The instrument was adapted from established frameworks in the literature and refined through expert consultation to ensure clarity and relevance to the context of SMEs.

### 3.3  Data analysis

Statistical analyses were performed using Python's Pandas and Statsmodels libraries. Descriptive statistics were calculated for each dimension and a correlation matrix was constructed to assess the interdependencies among them, identifying key parameters that influence the overall readiness of Industry 4.0. Regression analysis quantified the impact of each dimension, revealing that leadership is the most influential factor ($r = 0.92$) [10]. Furthermore, ANOVA was used to test for significant differences in readiness levels across sectors and regions, with p-values and confidence intervals ensuring statistical rigour.

### 3.4  Participant recruitment and ethical considerationsThe MITI recruited participants

through industry associations, chambers of commerce, and direct outreach. A pre-screening phase ensured eligibility based on firm size, industry, and geographic location. The eligible SMEs were then formally invited to participate, with detailed instructions provided to ensure data accuracy and comprehension.

Ethical approval was obtained in the accompanying guidelines, and informed consent was secured from all participants. Confidentiality and anonymity were maintained throughout the study and the voluntary nature of participation was emphasised.

### 3.5  Limitations

A limitation of this study is the potential bias due to variations in the digital infrastructure in different regions, which may affect the generalisability of the results. Additionally, reliance on self-reported data may introduce subjective biases, as responses reflect perceived rather than objective measures of readiness..

## 4  Results

The assessment of 506 Malaysian SMEs reveals significant variability in Industry 4.0 readiness in different dimensions, sectors, and regions. Leadership and strategic alignment emerge as relatively strong dimensions (see Table 1), with mean scores of 0.81 each, indicating that many SMEs have a strategic vision to adopt Industry 4.0 technologies. However, workforce competency was identified as the weakest dimension, with a mean score of 0.35, highlighting a critical gap in

**Table 1. Descriptive statistics of crucial dimensions.**

| Dimensions | Mean | Standard Deviation | Minimum | Maximum |
|---|---|---|---|---|
| Leadership | 0.81 | 1.00 | 0 | 4 |
| Collaboration Structure and Governance | 0.58 | 0.77 | 0 | 4 |
| Digital Infrastructure | 0.61 | 0.78 | 0 | 4 |
| Workforce Competency | 0.35 | 0.62 | 0 | 3 |
| Strategic Alignment | 0.81 | 1.04 | 0 | 4 |
| Overall readiness index | 0.81 | 0.70 | 0 | 4 |

the ability of SMEs to fully leverage advanced technologies [10]. The correlation analysis shows that leadership is the most vital factor, with a correlation coefficient of 0.92 with overall readiness, followed closely by governance (0.90) and digital infrastructure (0.89) [4,10].

Table 2 shows the correlation coefficients between the overall readiness index and other dimensions. The high correlation coefficients suggest that leadership, collaboration structure and governance, and digital infrastructure are critical factors influencing the overall readiness for Industry 4.0. Leadership has the highest correlation (0.92), indicating its paramount importance. This is consistent with the literature that highlights the importance of top management support and strategic alignment in successfully implementing Industry 4.0 initiatives [11,20]

The histogram in Fig 1, illustrates the distribution of the general readiness index among SMEs. The histogram reveals that most SMEs have a readiness index below 1, indicating low levels of preparedness. This trend towards lower readiness scores highlights the need for significant improvements across the board. Similar patterns have been observed in other studies on the readiness of SMEs for Industry 4.0, which often point to resource constraints and lack of expertise as significant barriers [10,25].

## 4.1 Sectoral analysis

Sector-specific differences reveal that the Chemicals and Chemical Products sector exhibits the highest readiness with a mean index of 1.10. On the contrary, sectors such as Fabricated Metal Products and Machinery and Equipment show lower readiness levels, with mean scores of 0.75 and 0.80, respectively. This disparity can be attributed to the varying degrees of technological advancement and digital infrastructure within these industries. Sectors like chemical industries benefit from robust technological investments and streamlined digital processes, which enable higher readiness levels than those with more manual or less digitised [8]. Fig 2 illustrates the variation of the readiness index in different sectors.

Furthermore, Table 3 presents the readiness index by section. The category "Others", including various smaller sectors, has the lowest mean readiness index of 0.60, reflecting a significant need for intervention and support.

## 4.2 Regional analysis

Geographically, Selangor and Johor lead in Industry 4.0 readiness, with mean indices of 1.20 and 1.10, respectively, driven by better digital infrastructure and supportive government policies. In contrast, Perak and smaller states exhibit lower readiness, with mean indices of 0.70 and 0.50, respectively, underscoring the uneven distribution of technological infrastructure and support across different regions [21,24]. Fig 3 and Table 4 show the differences in the readiness index in various states in Malaysia.

## 4.3 Interpretation of readiness scores

In this scoring system, a score of 1 indicates low readiness, while scores above 1 reflect progressively higher levels of preparedness to adopt Industry 4.0. The distribution of scores highlights the general need for improvements in the

**Table 2. Correlation coefficients with the readiness index.**

| Dimensions | Correlation with the readiness index |
|---|---|
| Leadership | 0.92 |
| Collaboration Structure and Governance | 0.90 |
| Digital Infrastructure | 0.89 |
| Workforce Competency | 0.85 |
| Strategic Alignment | 0.85 |

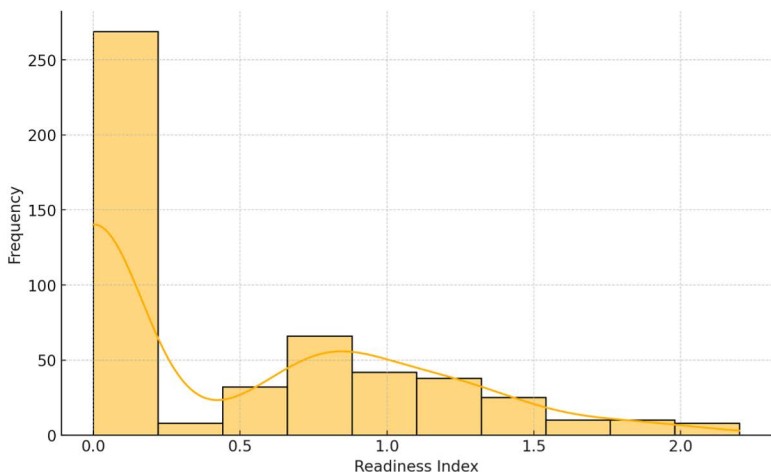

**Fig 1. Histogram of the readiness index.**

competency of the workforce and the digital infrastructure. These findings emphasise the importance of strategic leadership and targeted workforce development programmes to address skill gaps, particularly in underperforming sectors and regions.

## 5 Discussion

The findings of this study offer critical insight into the industry 4.0 readiness of Malaysian SMEs, providing a comprehensive understanding of the challenges and opportunities facing different sectors and regions. The strong correlation between leadership and general readiness underscores the crucial role of top management in driving digital transformation. This is consistent with the existing literature, which emphasises the importance of strategic leadership in successful Industry 4.0 adoption [11]. Strong leadership fosters the vision and alignment necessary for digital transformation, guiding SMEs through the integration of advanced technologies. However, this study also highlights significant gaps in workforce competency and digital infrastructure, which present substantial barriers to the full-scale implementation of Industry 4.0. These deficiencies are particularly pronounced in smaller sectors and less developed regions, where limited access to resources and support exacerbates the challenges of adopting advanced technologies [8].

The sectoral and regional disparities in this study highlight the need for targeted interventions. The Chemicals and Chemical Products sector and regions such as Selangor and Johor demonstrate comparatively high readiness levels, likely due to better access to digital infrastructure and proactive adoption of Industry 4.0 technologies. In contrast, sectors such as metal products and regions with less developed infrastructure exhibit a lower readiness, indicating that targeted interventions are needed to support these sectors and areas. This uneven distribution mirrors the broader challenges of

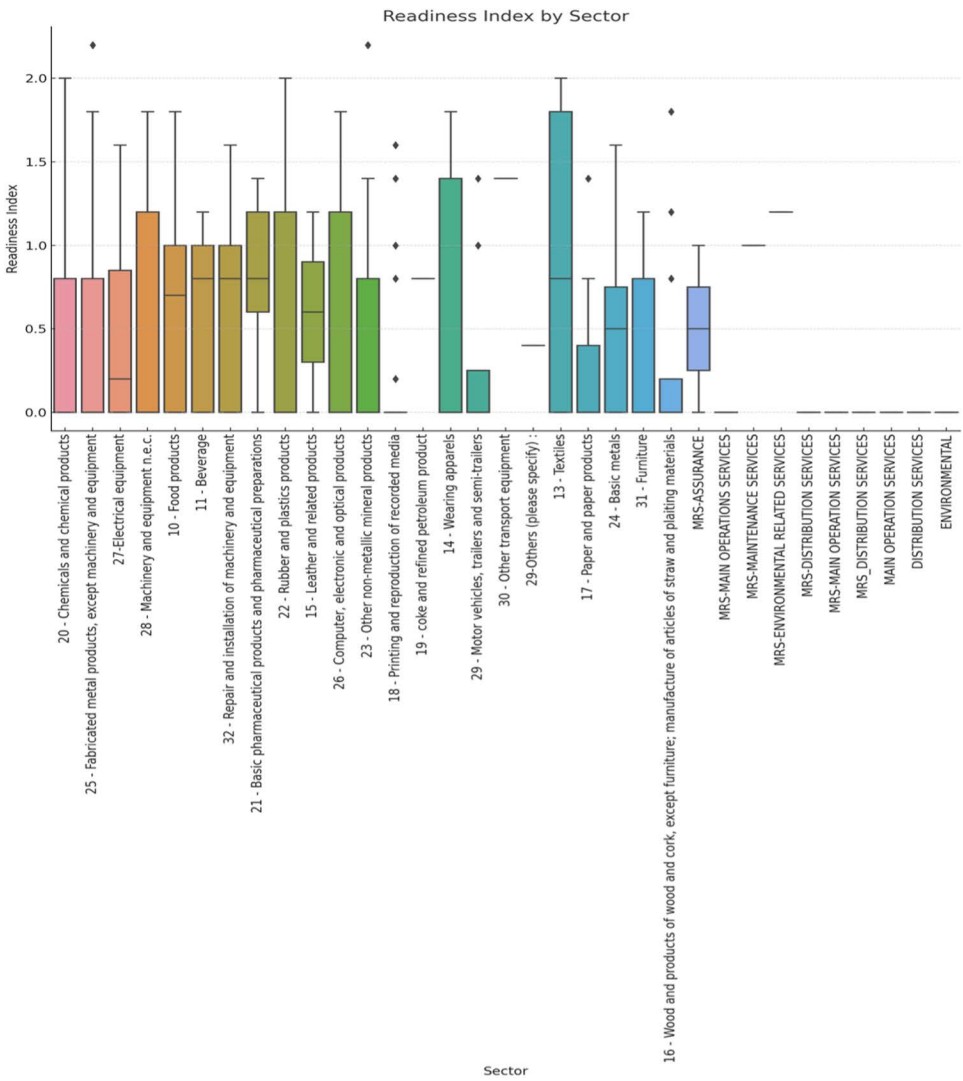

**Fig 2. Box plot of the Readiness Index by sector.**

**Table 3. Readiness index by sector.**

| Sector | Mean readiness index | Standard Deviation |
|---|---|---|
| Chemicals and Chemical Products | 1.10 | 0.90 |
| Electrical Equipment | 1.05 | 0.85 |
| Fabricated Metal Products | 0.75 | 0.70 |
| Machinery and Equipment | 0.80 | 0.75 |
| Others | 0.60 | 0.65 |

SMEs in other developing economies, where resource constraints and technological access disparities often hinder comprehensive digital transformation efforts [10]. The readiness levels across sectors highlight that Industry 4.0 adoption is uneven, with advanced sectors pushing ahead, while lagging sectors face challenges often rooted in resource constraints

**Fig 3. Box plot of the readiness index by state.**

**Table 4. Readiness index by state.**

| State | Mean readiness index readiness index | Standard Deviation |
|---|---|---|
| Selangor | 1.20 | 0.80 |
| Johor | 1.10 | 0.75 |
| Penang | 0.85 | 0.70 |
| Perak | 0.70 | 0.65 |
| Others | 0.50 | 0.60 |

and technological barriers. This mirrors regional disparities, where states like Selangor and Johor, with a better digital infrastructure, demonstrate higher levels of readiness.

In contrast, less developed states, such as Perak, lag with lower readiness scores. Investments in digital infrastructure, particularly in underdeveloped regions, are critical for bridging these gaps and ensuring national adoption of Industry 4.0 [6,21]. As developing economies increasingly integrate Industry 4.0 principles into their national agendas, the insights from this study provide a framework for policymakers to prioritise interventions that enhance digital infrastructure and workforce skills, thereby improving SME competitiveness on a global scale.

These insights are particularly relevant in the context of Malaysia's NIMP 2030, which aims to position Malaysia as a global industrial leader by 2030. NIMP 2030 emphasises the adoption of advanced technologies and innovation as critical drivers of economic growth, sustainability, and resilience. The plan highlights Industry 4.0 as a crucial component to modernising Malaysia's industrial sectors and improving its competitiveness on the global market. However, the findings of this study suggest that the success of NIMP 2030 will depend on addressing the critical gaps identified in the readiness of small businesses for Industry 4.0, particularly in leadership, workforce competence, and digital infrastructure.

As highlighted by this study and NIMP 2030, leadership development must be prioritised to ensure that top management in SMEs can drive the digital transformation process effectively. Strategic leadership will be vital in implementing the objectives of NIMP 2030, particularly in sectors that lag in industry 4.0 adoption. Government-led initiatives, such as executive leadership training and mentorship programmes, could be introduced to align SME leadership with the digital ambitions of the NIMP 2030 plan. Furthermore, as outlined in NIMP 2030, investments in digital infrastructure are essential to foster an enabling environment for Industry 4.0 technologies. This study identifies significant regional disparities in digital readiness, which mirror the greater national challenge of uneven technological development. To achieve the NIMP 2030 objectives, these infrastructure gaps must be addressed, focussing on less developed states such as Perak and others that need more digital readiness.

Workforce competency also emerges as a critical factor that could impact the success of NIMP 2030. The plan acknowledges that skilled labour will transform Malaysia's industrial landscape. However, this study highlights that workforce competency remains a significant challenge for many SMEs, particularly in the integration and use of advanced technologies. To address this, NIMP 2030's focus on human capital development must be implemented through comprehensive training and reskilling programmes that equip the workforce with the necessary skills to operate in an industry 4.0 environment. Collaboration between government, industry associations, and educational institutions will be essential to ensure that the workforce is adequately prepared for the digital future. Upskilling programmes, including certification courses and hands-on AI, IoT, and extensive data analytics training, could help close the workforce competency gap identified in this study.

Furthermore, the findings of this study align with the goal of NIMP 2030 of promoting inclusive and sustainable industrial growth. The readiness gaps between sectors and regions suggest that tailored sector-specific interventions must include every industry and region. Future research should address the limitations of this study, particularly the subjective bias inherent in self-reported data, which can affect perceived readiness levels. Additionally, longitudinal studies that track the progress of SMEs over time could offer valuable insight into the long-term impact of leadership and infrastructure improvements on readiness. Investigating readiness differences across SME subcategories, such as manufacturing vs. service-orientated firms, could also reveal specific challenges and needs, allowing more customised interventions. Finally, as digital transformation is a continuous process, future studies should explore the evolving readiness requirements of SMEs as new technologies emerge, helping to refine readiness frameworks and improve their applicability across diverse economic contexts.

### 5.1 Recommendations

Based on the findings, this study presents several recommendations to improve the readiness for Industry 4.0 of Malaysian SMEs, aligned with the objectives of the Industry4WRD programme and the Malaysian New Industrial Master Plan 2030 (NIMP 2030).

First, given the strong correlation between leadership and Industry 4.0 readiness, it is essential to support SME leaders through customised executive training programmes focused on digital transformation. Such programmes, as outlined in the Industry4WRD framework, should develop strategic leadership skills that align business objectives with the technological demands of Industry 4.0. Government agencies and industry associations could collaborate to offer these programmes, prioritising SMEs in sectors and regions with lower readiness levels.

Second, the significant disparities in digital infrastructure between regions suggest a need for targeted investments, particularly in states like Perak and others with limited technological resources. The Industry4WRD programme's

Intervention Fund, which provides financial support to improve digital tools and automation technologies, should prioritise funding for underdeveloped regions. Improving the digital infrastructure in these areas can help bridge regional readiness gaps, fostering an environment in which small businesses can more readily adopt and integrate advanced technologies.

Third, workforce competency has emerged as a critical area that needs improvement, as many SMEs lack skilled employees capable of leveraging Industry 4.0 tools effectively. We recommend implementing targeted training and reskilling initiatives to build the digital skillsets necessary for Industry 4.0 adoption, focussing on areas such as AI, IoT, and big data analytics. Government bodies, educational institutions, and industry stakeholders could partner to provide hands-on training, certification programmes, and internships that align with current industry demands.

Finally, given the sectoral differences observed in this study, a sector-specific approach within the Industry4WRD framework would better address the unique needs of different industries. For example, while the chemical sector could serve as a model for best practices, sectors such as fabricated metal products and machinery and equipment may require customised support to upgrade technology and access expert consultations. Continuous monitoring of SME progress through periodic assessments under the Industry4WRD framework can ensure that these interventions remain relevant and adapt as SME needs evolve.

## 6 Conclusions

This study provides a comprehensive assessment of the readiness of 506 Malaysian SMEs for Industry 4.0, identifying strengths and challenges in critical dimensions such as leadership, digital infrastructure, and workforce competency. The findings reveal that while leadership and strategic alignment are relatively well developed, workforce competency remains a critical gap, with significant implications for the ability of SMEs to successfully adopt and integrate Industry 4.0 technologies. The strong correlation between leadership and general readiness highlights the vital role of strategic leadership in fostering digital transformation, underscoring the need for targeted leadership development programmes. Sectoral and regional disparities further emphasise the uneven distribution of Industry 4.0 readiness, with advanced sectors such as chemicals and developed regions such as Selangor and Johor demonstrating higher preparedness. In contrast, sectors such as the Fabricated Metal Products and less developed regions require focused interventions to bridge these gaps.

These findings align with the goals of Malaysia's New Industrial Master Plan 2030 (NIMP 2030), which advocates inclusive growth and enhanced competitiveness through digitalisation. By addressing critical gaps in workforce skills, digital infrastructure, and leadership, policymakers and SME leaders can enhance SME readiness, ensuring that these enterprises are well equipped to navigate the demands of a digital economy. Future studies should explore longitudinal data to capture the progress of small businesses over time and industry-specific challenges that may affect readiness. As industry 4.0 technologies evolve, ongoing research and adaptive readiness frameworks will ensure that Malaysian SMEs remain competitive and resilient in a rapidly transforming global marketplace.

## Author contributions

**Conceptualization:** Muhammad Saqib Iqbal.

**Data curation:** Muhammad Saqib Iqbal.

**Formal analysis:** Muhammad Saqib Iqbal, Umawathy Techanamurthy.

**Funding acquisition:** Umawathy Techanamurthy.

**Methodology:** Zulhasni Abdul Rahim.

**Project administration:** Zulhasni Abdul Rahim.

**Software:** Umawathy Techanamurthy.

**Supervision:** Zulhasni Abdul Rahim.

**Validation:** Muhammad Saqib Iqbal.

**Writing – original draft:** Muhammad Saqib Iqbal.

**Writing – review & editing:** Muhammad Saqib Iqbal.

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
