## [Decision Letter · Decision Letter 0]

25 Oct 2024

PONE-D-24-42283Industry 4.0 Readiness and Strategic Plan Failures in SMEs: A Comprehensive AnalysisPLOS ONE

Dear Dr. IQBAL,

Thank you for submitting your manuscript to PLOS ONE. After careful consideration, we feel that it has merit but does not fully meet PLOS ONE’s publication criteria as it currently stands. Therefore, we invite you to submit a revised version of the manuscript that addresses the points raised during the review process.

We look forward to receiving your revised manuscript.

Kind regards,

Syed Hamid Hussain Madni

Academic Editor

PLOS ONE

**Journal Requirements:**

Reviewers' comments:

Reviewer's Responses to Questions

**Comments to the Author**

1. Is the manuscript technically sound, and do the data support the conclusions?

Reviewer #1: Yes

Reviewer #2: Partly

2. Has the statistical analysis been performed appropriately and rigorously? 

Reviewer #1: Yes

Reviewer #2: No

3. Have the authors made all data underlying the findings in their manuscript fully available?

Reviewer #1: Yes

Reviewer #2: No

4. Is the manuscript presented in an intelligible fashion and written in standard English?

Reviewer #1: Yes

Reviewer #2: Yes

5. Review Comments to the Author

**Reviewer #1:**  Abstract

1. The abstract lacks a clear introduction and problem statement. Please revise it to follow a structured format, including Background, Problem, Objective, Methods, and Results. Restructure the abstract to align with the standard format: Introduction, Problem Statement, Methodology, Results, and Conclusion.

Keywords

Please rewrite the keywords, ensuring that the first word is capitalized (e.g., Technology Adoption). Additionally, consider incorporating "Malaysia" or "Malaysian SMEs" to enhance searchability for region-specific research.

Introduction:

The author should elaborate on the relationship between NIMP 2030 and the Industry4WRD program earlier in this section.

Please include more recent statistics regarding the contribution of SMEs to Malaysia's GDP and employment.

Study Contribution, Research Questions and paper organization is missing in the introduction. Please ensure to:

1. Include a clear statement of the study's contribution and research question to the field.

2. Provide a brief overview of the paper's organization.

Literature Review:

The literature review provides a good foundation but could be further enhanced by:

1. Organizing the content into subsections, such as Industry 4.0 Concepts, SME Challenges, and Readiness Assessment Frameworks.

2. Including more recent references, particularly from 2024, to demonstrate the latest developments in the field.

3. The paper does not explicitly outline the research gap in the related works section. The author should compare the research gap with existing studies, providing a more critical analysis of the literature to highlight the gaps this study aims to fill. To write a clearer research gap in the related work section, the author may refer to the following manuscripts for guidance:

i. E-learning behavioral intention among college students: A comparative study. Education and Information Technologies, pp.1-23.

ii. Factors Influencing the Adoption of Industrial IoT for Manufacturing and Production SMEs in Developing Countries, IET Collaborative Intelligent Manufacturing, 6(1), p.e12093.

iii. IoT adoption model for e-learning in higher education institutes: a case study in Saudi Arabia. Sustainability, 15(12), p.9748.

iv. An empirical investigation of critical factors affecting acceptance of e-learning platforms: A learner’s perspective. SN Computer Science, 4(3), p.240.

v. Assessing the Prioritization of Key Influencing Factors for Industrial IoT Readiness in SMEs. International Conference of Reliable Information and Communication Technology. Cham: Springer Nature Switzerland, 2023.

Methodology:

The methodology is well-structured but could be improved by:

1. Providing more details on the sampling strategy (e.g., how the 506 SMEs were selected).

2. Explaining the rationale behind the 0-4 scale used in the readiness assessment.

3. Describing the specific statistical tests employed in the analysis, beyond just mentioning Python libraries.

4. Addressing potential limitations or biases in the data collection process.

Results:

The results section presents findings clearly but could be enhanced by:

1. Providing additional context for interpreting the readiness scores (e.g., what constitutes a "good" score?).

2. Including confidence intervals or p-values to support the statistical significance of the findings.

3. Explaining why certain sectors (e.g., Chemicals and Chemical Products) exhibit higher readiness.

Conclusion:

2. Discuss the broader implications of the findings for developing economies.

3. Suggest specific areas for future research based on the study's limitations or unanswered questions.

General Comments:

1. The paper would benefit from a thorough proofreading to address minor grammatical and typographical errors. Additionally, enhance the visual presentation of data by employing more varied and informative chart types.

2. Consider having the manuscript professionally proofread for language and clarity.

3. Verify that all figures are properly attached and referenced in the text.

4. Ensure consistent formatting of headings and subheadings throughout the paper.

**Reviewer #2: ** 1. Abstract Failed to demonstrate the background ,purpose, methodology, contribution of paper...need to rewrite

2. Introduction section lacks to show the need of this research work, Contribution of the paper etc.

3. Paper organization is missing in introduction section.

4. Literature review should discuss the current literature and critically analyse the literature, and show the need of your point of view in writing this research work.

5. The methodology section is incomplete. Does not discuss the method... Rather repeating the same literature review as of introductions action

6. Insufficient results and discussion.

6. PLOS authors have the option to publish the peer review history of their article (what does this mean? ). If published, this will include your full peer review and any attached files.

**Do you want your identity to be public for this peer review?** For information about this choice, including consent withdrawal, please see our Privacy Policy .

Reviewer #1: **Yes: ** Sajid Shah

Reviewer #2: No

---

## [Author Response · Author response to Decision Letter 1]

31 Oct 2024

Data availability statement: The data supporting this study's findings are available on request from the corresponding author, Iqbal. The data are not publicly available due to privacy and ethical restrictions.

For Reply to the reviewer's comments I have already attached the rebuttle document in the files section.

---

## [Decision Letter · Decision Letter 1]

11 Feb 2025

PONE-D-24-42283R1Industry 4.0 Readiness and Strategic Plan Failures in SMEs: A Comprehensive AnalysisPLOS ONE

Dear Dr. IQBAL,

Thank you for submitting your manuscript to PLOS ONE. After careful consideration, we feel that it has merit but does not fully meet PLOS ONE’s publication criteria as it currently stands. Therefore, we invite you to submit a revised version of the manuscript that addresses the points raised during the review process.

We look forward to receiving your revised manuscript.

Kind regards,

Sanmugam Annamalah

Academic Editor

PLOS ONE

Reviewers' comments:

Reviewer's Responses to Questions

**Comments to the Author**

1. If the authors have adequately addressed your comments raised in a previous round of review and you feel that this manuscript is now acceptable for publication, you may indicate that here to bypass the “Comments to the Author” section, enter your conflict of interest statement in the “Confidential to Editor” section, and submit your "Accept" recommendation.

Reviewer #2: (No Response)

2. Is the manuscript technically sound, and do the data support the conclusions?

Reviewer #2: Yes

3. Has the statistical analysis been performed appropriately and rigorously? 

Reviewer #2: Yes

4. Have the authors made all data underlying the findings in their manuscript fully available?

Reviewer #2: Yes

5. Is the manuscript presented in an intelligible fashion and written in standard English?

Reviewer #2: Yes

6. Review Comments to the Author

Reviewer #2: 1. Introduction section lacks to show the need of this research work. same text is repeated in methodology section and introduction section.

2. Literature review should discuss the current literature and critically analyse the literature.

3. The methodology section is incomplete. Does not discuss the method... need to describe the factors,whats items are included in data collection etc

7. PLOS authors have the option to publish the peer review history of their article (what does this mean? ). If published, this will include your full peer review and any attached files.

**Do you want your identity to be public for this peer review?** For information about this choice, including consent withdrawal, please see our Privacy Policy .

Reviewer #2: No

---

## [Author Response · Author response to Decision Letter 2]

14 Feb 2025

Response to Reviewer Comments

Dear Dr Annamalah and Reviewer 2,

We thank you for the thoughtful and constructive feedback on our manuscript, “Industry 4.0 Readiness and Strategic Plan Failures in SMEs: A Comprehensive Analysis.” We have carefully considered all comments and made revisions to improve the paper’s clarity, rigour, and overall quality. Below, we detail our responses to each of the reviewer’s comments:

Reviewer Comment 1:

“Introduction section fails to show the need for this research work. The same text is repeated in the methodology section and introduction section.”

Response:

We have restructured the Introduction to articulate the research gap and the necessity for assessing Industry 4.0 readiness among Malaysian SMEs. In the revised Introduction, we emphasise the unique challenges SMEs face and clearly state the contribution of our study. Additionally, we have removed any redundant methodological details previously appearing in the Introduction, ensuring that the Methodology section now exclusively covers the research design and data collection procedures.

Reviewer Comment 2:

“The manuscript must describe a technically sound scientific research piece with data supporting the conclusions. Experiments must have been conducted rigorously, with appropriate controls, replication, and sample sizes. The conclusions must be drawn appropriately based on the data presented.”

Response:

We affirm that the manuscript is technically sound and the data robustly supports our conclusions. Our study employed a rigorous experimental design with a sample of 506 SMEs, appropriate controls, and replication. To address this comment explicitly, we have added a new subheading in the Methodology section titled “Technical Rigor and Data Integrity.” In this subsection, we detail our survey instrument’s pilot testing and expert validation and the multiple statistical analyses (including correlation, regression, and ANOVA) performed to validate the relationships among key dimensions. These procedures ensure that our conclusions are drawn appropriately based on the data presented.

Reviewer Comment 3

“Literature review should discuss the current literature and critically analyse the literature.”

Response:

We have expanded the Literature Review section to include a more comprehensive discussion of recent studies. The revised literature review now critically analyses current findings, identifies existing gaps, and positions our research as a necessary contribution to the field of Industry 4.0 readiness. This expanded discussion clarifies how our multidimensional assessment addresses limitations in prior research and enhances our understanding of the challenges and opportunities for Malaysian SMEs.

Reviewer Comment 4:

“The methodology section is incomplete. It does not discuss the method. We need to describe the factors, what items are included in data collection, etc.”

Response:

We have extensively revised the Methodology section to include a detailed description of our data collection process and the measurement of key dimensions. Specifically, a new subheading, “Data Collection and Survey Instrument,” has been added. In this section, we describe the structured questionnaire designed to measure five key dimensions of Industry 4.0 readiness: leadership, governance, digital infrastructure, workforce competency, and strategic alignment. We explain that each item was rated on a standardised 0–4 Likert scale and provide examples of operationalising these constructs. This detailed description enhances the transparency and reproducibility of our study.

Additional Note:

The revisions made during the review process aim to clarify our methodology and emphasise the technical rigour of our research. These changes do not affect the underlying data, key findings, or the study's conclusions. Consequently, no modifications have been made to the abstract or findings.

We believe these revisions have significantly improved the clarity and quality of our manuscript. We thank you again for your valuable feedback and look forward to your favourable consideration of our revised submission.

Sincerely,

Muhammad Saqib Iqbal, PhD

Zulhasni Abdul Rahim, PhD

and Umawathy Techanamurthy, PhD

---

## [Decision Letter · Decision Letter 2]

21 Apr 2025

Industry 4.0 Readiness and Strategic Plan Failures in SMEs: A Comprehensive Analysis

PONE-D-24-42283R2

Dear Dr. Iqbal,

We’re pleased to inform you that your manuscript has been judged scientifically suitable for publication and will be formally accepted for publication once it meets all outstanding technical requirements.

Kind regards,

Ali Junaid Khan, PhD

Academic Editor

PLOS ONE

Additional Editor Comments (optional):

Reviewers' comments:

Reviewer's Responses to Questions

**Comments to the Author**

1. If the authors have adequately addressed your comments raised in a previous round of review and you feel that this manuscript is now acceptable for publication, you may indicate that here to bypass the “Comments to the Author” section, enter your conflict of interest statement in the “Confidential to Editor” section, and submit your "Accept" recommendation.

Reviewer #2: All comments have been addressed

2. Is the manuscript technically sound, and do the data support the conclusions?

Reviewer #2: Yes

3. Has the statistical analysis been performed appropriately and rigorously? 

Reviewer #2: Yes

4. Have the authors made all data underlying the findings in their manuscript fully available?

Reviewer #2: Yes

5. Is the manuscript presented in an intelligible fashion and written in standard English?

Reviewer #2: Yes

6. Review Comments to the Author

Reviewer #2: (Ghobakhloo, 2020a, b). There are two references in Bibliography (Ghobakhloo, 2020a, Ghobakhloo, 2020b). Need to Synchronize

7. PLOS authors have the option to publish the peer review history of their article (what does this mean? ). If published, this will include your full peer review and any attached files.

**Do you want your identity to be public for this peer review?** For information about this choice, including consent withdrawal, please see our Privacy Policy .

Reviewer #2: No

---

## [Editor Report · Acceptance letter]

PONE-D-24-42283R2

PLOS ONE

Dear Dr. Iqbal,

I'm pleased to inform you that your manuscript has been deemed suitable for publication in PLOS ONE. Congratulations! Your manuscript is now being handed over to our production team.

Kind regards,

on behalf of

Dr Ali Junaid Khan

Academic Editor

PLOS ONE